# Unravelling the Connection Between Energy Metabolism and Immune Senescence/Exhaustion in Patients with Myalgic Encephalomyelitis/Chronic Fatigue Syndrome

**DOI:** 10.3390/biom15030357

**Published:** 2025-03-01

**Authors:** Jente Van Campenhout, Yanthe Buntinx, Huan-Yu Xiong, Arne Wyns, Andrea Polli, Jo Nijs, Joeri L. Aerts, Thessa Laeremans, Jolien Hendrix

**Affiliations:** 1Pain in Motion Research Group (PAIN), Department of Physiotherapy, Human Physiology and Anatomy, Faculty of Physical Education and Physiotherapy, Vrije Universiteit Brussel (VUB), Laarbeeklaan 103, 1090 Brussels, Belgium; jente.van.campenhout@vub.be (J.V.C.); yanthe.buntinx@vub.be (Y.B.); huanyu.xiong@vub.be (H.-Y.X.); arne.wyns@vub.be (A.W.); jo.nijs@vub.be (J.N.); jolien.hendrix@vub.be (J.H.); 2Flanders Research Foundation-FWO, 1090 Brussels, Belgium; 3Department of Public Health and Primary Care, Centre for Environment & Health, KU Leuven, Kapucijnenvoer 35, 3000 Leuven, Belgium; 4Unit of Physiotherapy, Department of Health and Rehabilitation, Institute of Neuroscience and Physiology, Sahlgrenska Academy, University of Gothenburg, SE-405 30 Gothenburg, Sweden; 5Department of Physical Medicine and Physiotherapy, University Hospital Brussels, 1090 Brussels, Belgium; 6Laboratory of Neuro-Aging & Viro-Immunotherapy, Center for Neurosciences, Vrije Universiteit Brussel (VUB), Laarbeeklaan 103, 1090 Brussels, Belgium; joeri.aerts@vub.be (J.L.A.); thessa.laeremans@vub.be (T.L.)

**Keywords:** mitochondrial dysfunction, biomarker, natural killer cell, cytotoxic T cell, fatigue, pain

## Abstract

Myalgic Encephalomyelitis/Chronic Fatigue Syndrome (ME/CFS) is a debilitating disease, characterized by a diverse array of symptoms including post-exertional malaise (PEM), severe fatigue, and cognitive impairments, all of which drastically diminish the patients’ quality of life. Despite its impact, no curative treatments exist, largely due to the limited understanding of the disease’s underlying pathophysiology. Mitochondrial dysfunction, leading to impaired energy production and utilization, is believed to play a key role in the onset of fatigue and PEM, positioning it as a potential key pathophysiological mechanism underlying ME/CFS. Additionally, the disorder shows similarities to chronic viral infections, with frequent reports of immune system alterations, suggesting a critical role for immune (dys)functioning. In particular, the roles of immune senescence and immune exhaustion—two fundamental immune states—remain poorly understood in ME/CFS. This state-of-the-art review explores how metabolic dysfunction and immune dysfunction may be interconnected in ME/CFS, proposing that energy deficits may directly impair immune function. By examining this metabolic–immune interplay, this review highlights potential pathways for developing innovative therapeutic strategies that target both energy metabolism and immune regulation, offering hope for improving patient outcomes.

## 1. Introduction

Myalgic Encephalomyelitis/Chronic Fatigue Syndrome (ME/CFS) is a complex chronic condition affecting approximately 0.3% of the population worldwide, with a higher prevalence among women [1,2]. The debilitating disorder is characterized by a wide range of symptoms, including severe chronic fatigue, post-exertional malaise (PEM), widespread pain, and cognitive impairments, all of which drastically impact the patients’ quality of life and overall well-being [3]. The underlying pathophysiology of ME/CFS remains poorly understood, likely due to the large heterogeneity observed among patients [4]. Although biological differences in the immune, neuroendocrine, and autonomic systems have been reported in patients with ME/CFS, these research efforts have not yet led to a reliable and objective diagnostic test. As a result, the diagnosis of ME/CFS continues to rely solely on excluding all other medical causes and the clinical assessment of patients’ symptoms [2,3,4]. The lack of objective diagnostic tools not only adds to the burden for patients with ME/CFS but also highlights a critical unmet medical need in addressing this debilitating condition.

Although the exact pathophysiology of ME/CFS remains unclear, research has demonstrated dysregulation across multiple physiological systems, notably within energy metabolism and the immune system. Mitochondrial dysfunction, a critical factor affecting energy metabolism, is increasingly recognized as a driver of the energy deficits and hallmark symptoms of ME/CFS, including fatigue and PEM [5,6,7]. Evidence of mitochondrial impairments, such as reduced adenosine triphosphate (ATP) production rate, impaired oxidative phosphorylation (OXPHOS), and diminished spare respiratory capacity, has been observed in the cells of patients, leaving them unable to meet increased energy demands [5,8,9,10,11]. Moreover, compensatory metabolic shifts toward inefficient glycolysis and abnormal lactate production during low exertion further underscore the severity of metabolic dysfunction [8,12]. The immune system also appears to play a significant role in the pathophysiology of ME/CFS, as numerous cases have been associated with acute viral infections at disease onset [13,14]. However, it remains unclear whether an initial viral infection triggers a state of immune dysfunction or if a pre-existing, underlying immune dysfunction predisposes individuals to viral infections. In particular, immune senescence and immune exhaustion—two fundamental immune states—are well-recognized processes that might play a role in ME/CFS. Immune senescence involves a loss of replicative capacity of immune cells, whereas immune exhaustion pertains to the loss of functional cell activity required for immune protection [15]. Despite established significance in the immune system, research on their contribution to ME/CFS pathophysiology remains scarce. Importantly, energy metabolism and immune function are highly interconnected, with the potential to influence one another and create a feedback loop. This intricate relationship suggests that disturbances in one system may exacerbate dysfunction in the other, contributing to the complex and multifaceted nature of ME/CFS.

This state-of-the-art review provides comprehensive updates on energy metabolism, immune senescence, and immune exhaustion in ME/CFS. Moreover, it explores their interplay and collective role in the pathophysiology of ME/CFS, with energy deficits potentially impairing immune function. To enhance the readability of this review, Table 1 provides a description of the terms used.

## 2. Energy Metabolism

Energy metabolism is a fundamental process sustaining life by providing the energy required for cellular functions, relying on both aerobic and anaerobic pathways to meet cellular energy demands [16]. While anaerobic metabolism provides a rapid but limited energy supply, aerobic metabolism, primarily occurring in the mitochondria, is far more efficient and essential for sustained energy production [16]. Mitochondria have long been merely considered “powerhouses” of the cell due to their essential role in energy production, primarily through OXPHOS, which synthesizes ATP using the mitochondrial enzyme complex [17]. This tightly regulated process fuels cellular energy requirements, driving critical activities such as muscle contraction, biosynthesis, and iron transport [18]. In addition to ATP production, mitochondria are key producers of reactive oxygen species (ROS), a natural byproduct of OXPHOS [19,20]. At moderate levels, ROS play crucial roles as vital signalling molecules, functioning as second messengers in various cellular pathways. However, the excessive accumulation of ROS can induce oxidative damage and apoptosis, necessitating robust mitochondrial mechanisms to mitigate oxidative stress and maintain cellular homeostasis [21,22]. Such mitochondrial dysfunction has been implicated in ageing and a spectrum of diseases, including neurodegenerative disorders, cardiovascular conditions, and cancer [20,23,24,25,26]. Understanding the intricate balance between mitochondrial activity, ROS signalling, and oxidative stress provides crucial insights into cellular health and disease mechanisms. In the context of ME/CFS, mitochondrial dysfunction has emerged as a critical factor underlying the energy production deficits observed in patients, offering valuable insights into the disease’s pathophysiology.

### 2.1. Mitochondrial Dysfunction in ME/CFS

#### 2.1.1. ATP Production

A hallmark feature of mitochondrial dysfunction in ME/CFS is a reduction in the rate of ATP production [5,8]. This has been demonstrated in several studies using peripheral blood mononuclear cells (PBMCs) from patients with ME/CFS [10,11,27,28]. Mitochondrial dysfunction has therefore emerged as a plausible pathophysiological mechanism underlying ME/CFS [29,30]. The ATP profile test, a tool measuring ATP availability in neutrophils from a venous blood sample, has provided valuable insights into mitochondrial function in patients with ME/CFS (*n* = 138) compared to healthy controls (HCs, *n* = 53) [5,9]. Key parameters of mitochondrial respiration—such as basal respiration, ATP production, and maximal respiration—are significantly diminished [9]. Notably, the spare respiratory capacity, which reflects the ability of mitochondria to meet increased energy demands, is particularly impaired in severe cases of ME/CFS [10,11]. This reduced capacity highlights the inability of cells to adapt to heightened energy requirements, leaving them vulnerable to energy crises under stress or exertion and likely contributing to fatigue [11].

Despite the growing evidence, inconsistent findings related to mitochondrial OXPHOS remain. Vermeulen et al. reported normal OXPHOS capacity in PBMCs from patients with ME/CFS [28]. Their study also found that ME/CFS patients performed worse on exercise tests, suggesting that factors beyond mitochondrial enzyme complex activity may contribute to impaired ATP synthesis [28]. In contrast, Tomas et al. observed consistently reduced OXPHOS parameters in PBMCs from ME/CFS patients, particularly maximal respiration [10]. These conflicting results underline the complexity of mitochondrial dysfunction in ME/CFS and suggest that other mechanisms, including altered oxygen delivery or metabolic regulation, may play significant roles.

#### 2.1.2. Redox Imbalance and Energy Homeostasis

Redox imbalance has been identified as a fundamental characteristic of ME/CFS, indicating a disruption between ROS and antioxidant defences, causing oxidative stress [31]. Elevated oxidative stress, alongside reduced antioxidant capacity, has been observed in patients with ME/CFS and is thought to contribute to mitochondrial dysfunction [32,33]. This persistent redox imbalance is believed to underlie common symptoms of ME/CFS, such as fatigue, brain fog, and exercise intolerance [32]. The study by Jammes et al. classified patients with ME/CFS into two subgroups based on oxidative stress markers. The first subgroup exhibited higher oxidative stress markers and a reduction in antioxidant defences, while the second subgroup showed less pronounced oxidative stress but still differed significantly from healthy controls, with relatively better antioxidant capacity. This distinction further underscores the clinical heterogeneity observed in patients with ME/CFS [33]. In addition to oxidative stress, nitrosative stress, characterized by reactive nitrogen species (RNS), has also been reported in patients with ME/CFS [32].

Metabolic profiling of patients with ME/CFS provides additional evidence of mitochondrial dysfunction. Elevated markers of oxidative stress suggest increased cellular damage due to ROS [34]. In parallel, proteomic studies using SWATH-MS analysis have revealed differentially expressed proteins in PBMCs from patients with ME/CFS. These changes highlight impairments in oxidative phosphorylation, redox regulation, and mitochondrial function, collectively supporting deficient ATP production and increased oxidative stress in ME/CFS [35].

Further insights into mitochondrial dysfunction have been derived from lymphoblastoid cell lines from patients with ME/CFS. These cells demonstrate a striking inefficiency in ATP synthesis by Complex V, which impairs the final step of ATP production, accompanied by a compensatory upregulation of glycolysis [8]. This metabolic shift reflects an adaptive mechanism to partially offset deficient mitochondrial energy production, albeit with less efficient energy output. The systemic implications of these cellular abnormalities are evident in exercise-related studies [12].

Also, the regulators of energy homeostasis have been found to be deviant in patients with ME/CFS. The protein kinase Target of Rapamycin Complex I (TORC1) is a key regulator of cellular metabolism and energy homeostasis, playing a significant role in mitochondrial function, particularly within the OXPHOS complexes [36]. Lymphoblasts derived from patients with ME/CFS demonstrate abnormalities in mitochondrial respiratory function and elevated TORC1 activity. TORC1 facilitates the synthesis of mitochondrial proteins by promoting the translation of nuclear-encoded mitochondrial enzymes, leading to a specific upregulation of subunits in Complex I and Complex V [8]. These findings suggest that ATP production, specifically in Complex V, is impaired in lymphoblasts of patients with ME/CFS [8].

### 2.2. Links Between a Dysregulated Energy Metabolism and Clinical Symptoms

Studies suggest a strong correlation between mitochondrial dysfunction and the severity of ME/CFS symptoms [5,37]. Fatigue is a prominent symptom in patients with ME/CFS and is frequently linked to mitochondrial dysfunction [6]. However, the pathophysiological mechanisms underlying this association remain poorly understood.

One proposed biomarker for fatigue in ME/CFS is dysregulated carnitine levels, which is hypothesized to contribute to mitochondrial inefficiency [6]. Carnitine is essential for fatty acid oxidation and energy metabolism. Additionally, carnitine supports mitochondrial integrity and reduces the production of ROS, underscoring its role in maintaining cellular energy balance [38]. Another mitochondrial biomarker implicated in fatigue is Coenzyme Q10 (CoQ10) [6]. This critical component of the electron transport chain plays a key role in ATP production and antioxidant defence. Studies have reported significantly lower plasma CoQ10 levels in patients with ME/CFS compared to HCs, suggesting that CoQ10 deficiency is a significant contributor to the pathophysiology of ME/CFS [39,40,41,42].

PEM, another defining feature of ME/CFS, may also stem from systemic mitochondrial and metabolic dysfunction. Evidence points to increased post-exercise metabolic activity in patients with ME/CFS, leading to a hypermetabolic state. This is accompanied by reduced acetylation levels, a crucial cellular process for energy production and gene regulation, as well as dysregulated purine metabolism, which impairs energy availability. Collectively, these disruptions exacerbate the energy deficit observed in ME/CFS and contribute to the persistence of PEM [7]. Moreover, ME/CFS patients exhibit abnormal blood lactate responses during physical activity, coupled with impaired recovery after exertion. Even at low-to-moderate exercise intensities, patients show an earlier shift to anaerobic metabolism [12,34,43,44]. As the ATP synthesis is mostly decreased, anaerobic glycolysis in muscle needs to produce the extra ATP needed for the exercise, reflecting the lactate production [27].

Elevated levels of exosome-associated mitochondrial DNA (mtDNA) have been detected in patients with ME/CFS and correlated with the severity of ME/CFS symptoms [45,46]. However, no disease-causing mutations in mtDNA have been identified in this patient population [47]. The study by Venter et al. found mildly deleterious variants in mtDNA in both patients and HCs. However, as these may not be specific to ME/CFS, the presence of these variants in mtDNA could still contribute to mitochondrial dysfunction in some patients [48].

While the exact role of mitochondrial dysfunction in ME/CFS pathogenesis is not fully understood, evidence strongly suggests that it contributes to the pervasive energy deficits experienced by patients (see Figure 1 for a summary of findings presented in this review). Variability in study methodologies, patient cohorts, and diagnostic criteria may have led to conflicting findings, underscoring the need for standardized protocols in future research. Establishing uniform diagnostic and analytical methods will be critical to resolving these inconsistencies and fully elucidating the role of mitochondrial dysfunction in ME/CFS [9]. As research progresses, understanding these metabolic abnormalities may open new avenues for diagnosis, treatment, and symptom management in this debilitating condition. In the next section, we will dig into the immune system of patients with ME/CFS. From there, we will explain how metabolic dysfunction and immune dysfunction may be linked in patients with ME/CFS.

### 2.3. Potential Regulatory Mechanisms Underlying a Dysregulated Energy Metabolism

While disturbances in energy metabolism have been widely documented in ME/CFS, the regulatory mechanisms driving these abnormalities remain poorly understood. Epigenetic modifications, particularly DNA methylation, have gained increasing attention as potential contributors to these metabolic abnormalities in ME/CFS [49,50]. DNA methylation is a dynamic mechanism that can influence gene expression without altering the genetic code. Recent advances have shed light on the complex interplay between energy metabolism and DNA methylation, revealing a bidirectional relationship in which metabolic changes can influence DNA methylation, while epigenetic modifications can, in turn, regulate metabolic pathways [51]. This dynamic interaction has been implicated in various diseases, including metabolic disorders, cancer, and neurodegenerative conditions, suggesting that disruptions in this balance may contribute to disease pathogenesis [52]. Two separate studies identified alterations in DNA methylation patterns in the PBMCs of patients with ME/CFS, mainly affecting cellular metabolism and immune signalling [49,50]. These findings suggest that the metabolic environment may play a crucial role in shaping the epigenetic landscape in ME/CFS, potentially leading to persistent cellular dysfunction.

## 3. Immune Dysfunction in ME/CFS

The immune system plays a critical role in defending the body against pathogens and injury, relying on a complex network of physical barriers, immune cells, proteins, and signalling molecules to coordinate an effective immune response [53]. The dysregulation of the immune system has been linked to a wide spectrum of diseases such as cancer, autoimmune diseases, and neurodegenerative disorders [53,54,55]. While the precise role of the immune system in ME/CFS remains elusive, immunologic disturbances have been associated with symptom severity in affected individuals [56,57]. Studies have identified several immune abnormalities in patients with ME/CFS, including low-grade inflammation, oxidative stress, cytokine imbalances, impaired T and B cell functionality, dysfunctional antiviral enzymes, and diminished natural killer (NK) cell activity [3,58,59,60,61,62,63]. Despite the identification of numerous immune abnormalities, results are often inconsistent between studies [61,62,63]. A limitation in current research is the lack of targeted investigation into specific immune functions. Instead, many studies adopt a broad approach, often with the aim of identifying patterns or abnormalities. Nonetheless, these studies on ME/CFS remain valuable, particularly when their results can be interpreted in the context of specific immune functions such as immune senescence and immune exhaustion—two processes that impair immune cell function through distinct molecular pathways [15].

### 3.1. Immune Senescence

#### 3.1.1. What Is Immune Senescence?

Senescence, also known as cellular ageing, refers to a cell state of irreversible cell cycle arrest triggered by ageing or stress [64]. While senescence plays beneficial roles in processes such as wound healing and tumour suppression, chronic senescence induced by stress stimuli such as oxidative stress, persistent DNA damage, and activated oncogenes can lead to pathological outcomes [64,65]. In addition to growth arrest, senescent cells exhibit other hallmark features, including critical telomere shortening, distinct transcriptional and epigenetic profiles, resistance to apoptosis, and the senescence-associated secretory phenotype (SASP) [65,66,67,68]. SASP involves the release of signalling molecules such as interleukins, chemokines, and growth factors, including interleukin 6 (IL-6) and interferon gamma (IFN-γ) [68].

In the immune system, senescence diminishes the replicative capacity of cell populations like T cells, B cells, and NK cells [15]. Immune senescence is associated with functional impairments, including reduced responsiveness to new antigens, impaired memory T cell responses, increased autoimmune susceptibility, and systemic, chronic low-grade inflammation [69]. Emerging evidence links immune senescence to lifestyle factors such as sleep disturbances (e.g., chronic insomnia) and chronic stress, both of which are prominent features of ME/CFS [3,70,71]. Given the interplay between chronic stress, chronic insomnia, and immune dysfunction, these observations provide a starting point for exploring the pathophysiology of ME/CFS.

Immune cells are tightly regulated by stimulatory and inhibitory receptors, which undergo dynamic changes during differentiation [15]. Upon binding with their ligands, these receptors regulate cell activation and function by modulating signalling pathways. The level of receptor expression therefore finetunes the immune response [72,73]. For instance, stimulatory receptors such as CD27 and CD28 are prominently expressed in the early differentiation stages, while their expression progressively decreases as T cells mature and potentially acquire a senescent phenotype [15]. Senescent T cells are characterized by the overexpression of inhibitory receptors, including CD57, killer cell lectin-like receptor G1 (KLRG-1), and T-cell immunoglobulin and mucin domain 3 (TIM-3), alongside a complete loss of CD27 and CD28 expression [15,72]. In some cases, senescent T cells also upregulate receptors such as killer cell immunoglobulin-like receptors (KIR) and natural killer group 2A (NKG2A), although these are not conventionally used as markers of T cell senescence [74,75]. Senescent T cells also shift from naïve to memory phenotypes and produce more proinflammatory cytokines such as IL-6, IFN-γ, and tumour necrosis factor alpha (TNF-α) [76,77]. An overview of immune senescence features can be found in Figure 2.

Immune senescence is also apparent in B cells of aged individuals, highlighting the association between senescence and ageing [78]. B cell senescence is marked by a shift in the balance between memory and naïve B cell populations, reducing their ability to respond to novel pathogens—a hallmark of immune senescence [78]. The increased prevalence of memory B cells contributes to elevated levels of proinflammatory cytokines such as TNF-α [79,80]. While CD27 downregulation is commonly used to distinguish memory from naïve B cells, its role in identifying senescent B cells requires further investigation [81].

Unlike T and B cells, the number of NK cells do not decline in number with age. Instead, the NK cell population expands, altering cytokine expression and immune function [82]. It remains uncertain whether senescent NK cells share characteristics with senescent T cells, as there are currently no established markers for NK cell senescence. Nonetheless, changes in the expression of receptors including NKG2A, CD57, and KIR suggest a reduced proliferative capacity in NK cells [83,84]. The receptor KLRG-1, known to reduce proliferative capacity in T cells, similarly affects about 50% of NK cells, hinting at potential parallels in senescence-related mechanisms [85].

#### 3.1.2. Immune Senescence in ME/CFS

To date, no study has extensively studied immune senescence within the context of ME/CFS. While premature telomere attrition—a hallmark of senescence associated with accelerated ageing—has been observed in patients with ME/CFS, this finding was not specific to immune cells [86]. Cytokine dysregulation, another feature associated with immune dysfunction, has been observed in ME/CFS but remains inconsistent among studies. Corbitt et al. reviewed the current understanding of cytokine patterns in ME/CFS, noting inconsistency and a lack of significance across studies [62]. For instance, IL-6 has been reported as both elevated and decreased in different ME/CFS cohorts [87,88,89,90,91]. It is important to note that different ME/CFS diagnostic criteria and analytical methods are available, which is likely to influence the variability across studies and complicate direct comparisons [62]. The overall variability makes it difficult to use cytokine dysregulation as a reliable marker of immune senescence. Nevertheless, associations between ME/CFS symptoms and cytokines may hold clinical relevance. Montoya et al. found a positive correlation between symptom severity and certain cytokines, including IFN-γ, even though no significant differences between patients and HCs were identified [92]. Decreased IFN-γ mRNA expression has been observed in NK cells from patients with ME/CFS, while T cells did not show a significant reduction [89]. Paradoxically, the T cells of patients with ME/CFS exhibited an increased production of IFN-γ and TNF-α in this same study, underscoring the complexity of immune alterations in ME/CFS [89].

Studies into surface markers provide further insights but also reveal inconsistencies. While the increased expression of KIR and the reduced expression of CD57 and KLRG-1 on T cells have been reported, these findings are not consistently replicated across ME/CFS studies [93,94]. For instance, Curriu et al. found no differences in CD57, CD27, or CD28 expression, suggesting that T cell senescence may not be a dominant feature of ME/CFS [95]. Similarly, research on NK cells has not revealed significant changes in potential senescence markers such as CD57 and NKG2A [95,96,97,98]. These results could indicate that CD57 is not a representative marker of NK cell senescence or that senescent NK cells might not be present in ME/CFS. This highlights the need for more research into the robust markers of NK cell senescence, specifically within the context of ME/CFS. Findings related to B cell senescence are even more limited, with no observed differences in CD27 expression in patients with ME/CFS, which could be a potential senescence marker but was used to discriminate between naïve and memory B cell populations, leaving the role of senescence in B cells unclear [95,99].

Although current evidence does not strongly indicate immune senescence in ME/CFS, some preliminary findings, such as elevated KIR expression, warrant further research. Future studies with larger, well-characterized cohorts are needed to clarify the potential role of immune senescence in ME/CFS pathophysiology. Prior research on immune senescence in cancer, autoimmune diseases, and ageing could provide a valuable framework for exploring its role in ME/CFS.

### 3.2. Immune Exhaustion

#### 3.2.1. What Is Immune Exhaustion?

Immune exhaustion is a distinct form of dysfunction characterized by a progressive loss of functional activity in lymphocytes, particularly T cells [15]. Unlike senescence, which arises from extensive replication, immune exhaustion is driven by sustained antigenic stimulation and prolonged immune activation, such as during chronic viral infections or cancer [15,100,101]. An important distinction between exhaustion and senescence is the reversible nature of exhaustion, in contrast to the more terminal state of senescence [15]. This underscores the importance of the identification of immune exhaustion and differential diagnosis with immune senescence in patients with ME/CFS.

The key features of immune exhaustion include the loss of effector function, overexpression of inhibitory receptors, metabolic dysregulation, and both transcriptional and epigenetic changes distinct from those associated with senescence [102,103,104]. While most research has focused on T cells, emerging evidence suggests that exhaustion may also occur in B cells and NK cells [105]. T cell exhaustion is typically identified by the overexpression of inhibitory receptors such as programmed cell death protein 1 (PD-1), cytotoxic T-lymphocyte–associated antigen 4 (CTLA-4), TIM-3, and T-cell immunoreceptor with Ig and ITIM domains (TIGIT), 2B4 (also known as CD244), and lymphocyte-activation gene 3 (LAG-3) [100,106]. The co-expression of receptors, such as galectin-9 (Gal-9) with PD-1 or TIGIT, works synergistically and reinforces T cell exhaustion [107]. While exhaustion and senescence share some similarities—such as reduced functionality and the altered expression of surface receptors—specific combinations and the level of receptors help differentiate the two. Additionally, transcription factors involved in T cell development and differentiation such as T-box expressed in T-cells (T-bet), eomesodermin (EOMES), and T cell factor 1 (TCF-1) are dysregulated in exhausted T cells [100,104,108]. This is accompanied by decreased cytokine production, including interleukin 2 (IL-2), IFN-γ, and TNF-α, while increased levels of interleukin 10 (IL-10) have been implicated in promoting T cell exhaustion [100,104]. These characteristics of immune exhaustion are depicted in Figure 2.

Although exhaustion is well characterized in T cells, its occurrence in other immune cells is less clear. Impaired NK cell functionality is associated with reduced effector function and secretion of IFN-γ and TNF-α, though it is still unclear whether this can be defined as immune exhaustion [103,109]. Exhausted-like NK cells express both stimulatory and inhibitory receptors, many of which overlap with T cell exhaustion, such as PD-1, TIM-3, TIGIT, LAG-3, and NKG2A, as well as activating receptors such as natural killer group 2D (NKG2D) and DNAX accessory molecule 1 (DNAM-1) [103,109,110,111,112,113]. However, studies have reported inconsistent results regarding the expression levels of exhaustion-related receptors such as TIM-3, leaving uncertainty about whether and how exhaustion is present in NK cells [111,114]. The significantly reduced expression of T-bet and EOMES in these exhausted-like NK cells further supports functional impairment [115]. Similarly, B cell exhaustion has been described but remains less explored compared to T cell exhaustion. The potential exhausted states of NK and B cells were previously explained in a comprehensive overview by Kevin Roe [105].

#### 3.2.2. Immune Exhaustion in ME/CFS

Only a few studies have investigated immune exhaustion in ME/CFS, but emerging evidence suggests potential involvement. A recent transcriptomic study revealed abnormalities in T and NK cells consistent with exhaustion, including the downregulation of both type 1 and type 2 IFN signalling and reduced IFN-γ production [116]. These findings align with other studies reporting diminished levels of IFN-γ and TNF-α, indicative of an impaired and potentially exhausted T cell state [117,118,119]. However, contrasting findings, such as elevated levels of IFN-γ and TNF-α in patients with Long COVID and ME/CFS symptoms, highlight the complexity and variability of immune alterations in ME/CFS [88]. Elevated levels of TNF-α and IL-10 have been correlated with ME/CFS symptoms and disease severity, but inconsistencies across studies limit their utility as reliable biomarkers [57,62,120]. Iu et al. recently provided evidence of T cell exhaustion in ME/CFS, identifying epigenetic features such as reduced chromatin accessibility in genes associated with Nuclear factor kappa B (NF-κB)/TNF-α signalling [121]. Similarly, Walitt et al. reported the differential expression of genes within the IL-10 and NF-κB pathways in male patients with ME/CFS, with a trend toward IL-10 upregulation and NF-κB downregulation, indicative of an exhausted T cell phenotype [122]. Additionally, Iu et al. observed the upregulation of T-bet, EOMES, and TCF-1 in affected individuals, further supporting the presence of immune exhaustion in ME/CFS [121].

Investigations into cell surface receptor expression further support a role for immune exhaustion in ME/CFS. Eaton-Fitch et al. reported pathways related to PD-1 and CTLA-4 signalling to be upregulated in the T cells of patients with ME/CFS [116]. This finding is in line with another study reporting the upregulation of PD-1 and CD95 on CD4^+^ and CD8^+^ T cells, respectively [95]. In contrast, Iu et al. did not report significant changes in PD-1 expression between patients with ME/CFS and HCs, even though they did observe an elevated PD-1 trend in patients [121]. Adding to this, Walitt et al. reported increased PD-1 expression on CD8^+^ T cells in the cerebrospinal fluid (CSF) of patients with ME/CFS, whereas PD-1 levels on CD4^+^ T cells remained unchanged in both blood and CSF [122]. TIGIT and CD244 expression showed no significant alterations in their ME/CFS cohort either, while the expression of CD266—likely to be downregulated in T cell exhaustion—was decreased [101,122]. Supporting the involvement of T cell exhaustion, Saito et al. showed that patients with Long COVID and ME/CFS symptoms have elevated expressions of PD-1, TIM-3, TIGIT, and Gal-9 [88]. Furthermore, they report TIM-3^+^ cells to be associated with impaired TNF-α and IFN-γ expressions, whereas PD-1^+^ cells correlated with an elevated expression of these cytokines [88]. Interestingly, Steiner et al. demonstrated an association between ME/CFS and an autoimmune risk allele variant of CTLA-4, suggesting a genetic predisposition to immune exhaustion in these patients [123].

Despite these findings, no definitive correlation has been established between immune exhaustion and ME/CFS symptoms, leaving significant gaps in our understanding, especially regarding its clinical importance. The heterogeneity of ME/CFS and the inconsistencies across studies further underscore the complexity of elucidating the role of immune dysfunction in this disorder. More comprehensive studies investigating immune exhaustion are needed to unravel its role in ME/CFS pathophysiology. Current insights from research on cancer and human immunodeficiency virus (HIV) provide a valuable framework for exploring how immune exhaustion might contribute to immune impairment in ME/CFS [101,124]. Considering the reversible nature of immune exhaustion highlighted above, the discovery of an immune checkpoint blockade such as CTLA-4, which revolutionized cancer therapy, underscores the potential of targeting exhaustion in patients with ME/CFS [125].

## 4. Exploring the Metabolic–Immune Link in ME/CFS

### 4.1. Metabolic–Immune Interplay

Research has continuously highlighted the critical role of mitochondrial function and energy metabolism in immune responses [126,127,128,129]. Mitochondria regulate immune cell activation, differentiation, and survival, while driving immune responses through the release of mitochondrial damage-associated molecular patterns (DAMPs) such as circulating mtDNA [126,127,128]. Additionally, mitochondria contribute to innate immunity by serving as signalling platforms for antiviral molecules, including mitochondrial antiviral signalling proteins (MAVS) [127]. A key feature of immune activation is the metabolic shift from OXPHOS and fatty oxidation to aerobic glycolysis and glutamine metabolism [130,131]. This metabolic transition enables immune cells to meet the bioenergetic demands associated with differentiation and effector functions. It is tightly regulated by a variety of signalling pathways to ensure an appropriate immune response. Consequently, the dysfunction of these mitochondrial processes and the dysregulation of energy homeostasis can impair immune cell function, leading to immunodeficiencies and increased susceptibility to infections [129].

The immune system also plays a significant role in shaping mitochondrial function and energy metabolism. Chronic immune activation and prolonged antigenic exposure further disrupt mitochondrial function and energy metabolism, leading to metabolic dysregulations [132]. Persistent antigenic exposure triggers inhibitory signalling pathways, including PD-1 and CTLA-4, disrupting metabolic pathways and leading to mitochondrial depolarization, defective OXPHOS, and increased ROS production [133,134,135]. The resulting oxidative stress can damage mtDNA, exacerbating cell dysfunction, and ultimately resulting in energy deficits that promote immune exhaustion [135]. Immune senescence is also closely associated with mitochondrial dysfunction and a switch in energy metabolism. Specifically, senescent T cells exhibit impaired mitochondrial function, a reliance on glycolysis, and elevated ROS production [136]. Notably, distinct patterns of metabolic dysregulation emerge among different T cell types, with mitochondrial mass playing a defining role [137]. For instance, CD4^+^ T cells exhibit a higher mitochondrial mass, enabling the increased uptake of glucose and lipids compared to CD8^+^ cells. This metabolic advantage in CD4^+^ T cells facilitates enhanced proliferation and reduces susceptibility to senescence [137].

The relationship between mitochondrial dysfunction and immune activation has been recognized as a bidirectional feedback loop as observed in ageing and disease contexts, including Parkinson’s disease [138]. This interplay exacerbates disease symptoms and contributes to a self-perpetuating cycle of dysfunction [139]. Emerging data suggest that altered energy metabolism and immune dysfunction may be closely linked in ME/CFS [8,27,59], as highlighted in the next section.

### 4.2. Metabolic–Immune Alterations in ME/CFS

Given the involvement of mitochondrial dysfunction in both immune senescence and exhaustion, findings highlight a potential interplay between these processes in ME/CFS. By integrating immunometabolism into ME/CFS research, novel disease mechanisms may be uncovered, providing potential therapeutic targets. This review summarizes current knowledge on immunometabolism in ME/CFS and hypothesizes that mitochondrial dysfunction underlies immune senescence and/or exhaustion in these patients, reinforcing the interconnected nature of mitochondrial dysfunction and immune dysregulation in ME/CFS.

Mitochondrial dysregulation plays a critical role in modulating immune function and may significantly contribute to the pathophysiology of ME/CFS. mTORC1, whose activity was found to be elevated in patients with ME/CFS, might be a key regulator in this process, as its dysregulation can disrupt cellular energy production and thus affect immune cell function [8,140]. An additional mechanism linking mitochondrial dysfunction to immune dysregulation involves the release of mtDNA. As previously mentioned, circulating mtDNA functions as a DAMP, which can activate the immune system and drive chronic inflammation. This observation aligns with the persistent inflammatory state often observed in patients with ME/CFS [47]. Moreover, the mtDNA of patients with ME/CFS has been shown to stimulate human microglia—key immune cells within the central nervous system—in vitro [46]. This stimulation triggers the release of proinflammatory cytokines, particularly IL-1β, which is a central mediator of neuroinflammation. Although studies exploring this in humans with ME/CFS are needed, these findings suggest a mechanism in which mitochondrial stress in peripheral cells signals to the central nervous system, contributing to chronic (neuro)inflammation and the neurological symptoms of ME/CFS [46]. The increased production of ROS, driven by inflammation, is another contributing element of mitochondrial dysregulation in ME/CFS. This increased production of ROS is a primary contributor to oxidative stress, which contributes to cellular damage and exacerbates mitochondrial dysfunction, creating a feedback loop that further amplifies inflammation and immune dysregulation [32].

Metabolic and mitochondrial dysfunction in immune cells could provide insights into the possibility of immune senescence and exhaustion in ME/CFS pathophysiology. Patients with ME/CFS exhibit reduced glycolysis in both CD4^+^ and CD8^+^ T cell subsets, consistent with a state of hypometabolism [59]. Hypometabolism may indicate a reduced capacity for accurate immune responses, aligning with features of immune exhaustion. A recent study showed that the T cells of patients with ME/CFS increasingly rely on fatty acid oxidation (FAO) for ATP production, have reduced glycolysis, and increased uptake of exogenous fatty acids, potentially impacting immune regulation [141]. These metabolic adaptations resemble patterns seen in T cell exhaustion during chronic infections and cancer, further supporting the hypothesis of immune exhaustion in ME/CFS [141]. Furthermore, CD8^+^ T cells in ME/CFS show reduced mitochondrial membrane potential, suggesting mitochondrial dysfunction, which may in turn be associated with altered cytokine profiles [59]. Mandarano et al. reported negative correlations between resting glycolysis and cytokines such as IL-2 and TNF-α, uniquely observed in patients with ME/CFS [59]. This finding links proinflammatory cytokines to reduced metabolic activity, suggesting that chronic inflammation in ME/CFS impairs energy metabolism in immune cells.

As Hornig explained, the patterns of metabolic dysfunction are only partially explained by immune senescence and/or exhaustion and require further research [142]. Metabolic and mitochondrial dysregulation in the T cells of patients with ME/CFS may involve deficits in basal glycolysis, decreased mitochondrial proton leak, reduced ATP production, decreased membrane potential, and increased mitochondrial mass [142]. The observed metabolic and mitochondrial dysregulations extend to other immune cells as well. NK cells in ME/CFS show a decrease in glycolytic reserve, limiting their ability to increase ATP production during immune responses [143]. Lipid metabolism alterations may further contribute to NK cell dysfunction in ME/CFS. Recent evidence shows reduced levels of CD36, a key fatty acid transporter, combined with increased FAO in patients with ME/CFS, suggesting a compensatory mechanism to prevent lipid overload [141]. Excessive fatty acids may suppress mTOR activity, and given that mTORC1 activity is already dysregulated in ME/CFS, this additional metabolic impairment may further reduce NK cell function [141]. The B cells of patients with ME/CFS demonstrate a reduced mitochondrial mass and increase their use of amino acids upon stimulation. This indicates a reliance on alternative substrates to maintain ATP levels, providing evidence for a dysregulated energy metabolism [144]. This variation in energy metabolism might be associated with CD24 expression in B cells but requires further research [144,145]. One study measured OXPHOS in PBMCs, not distinguishing between different immune cell types, and noted normal glycolysis [10]. They also revealed reduced mitochondrial respiration, which is a feature potentially linked to immune exhaustion. Figure 3 presents a schematic overview of the interaction between energy metabolism and the immune system, illustrating how disruptions in these systems may exacerbate each other in ME/CFS.

The bidirectional feedback loop between mitochondrial dysfunction and immune senescence/exhaustion in ME/CFS underscores the complex relationship between immune regulation and metabolism. Dysregulations in the key pathways of mitochondria described in this review underscore the metabolic impairments that characterize immune cells in patients with ME/CFS. Conversely, immune exhaustion could further impair mitochondrial dysfunction. However, the patterns of metabolic dysfunction in ME/CFS remain only partially explained by immune senescence and exhaustion, necessitating more detailed investigations. The further exploration of these immune–metabolic disruptions could unveil novel insights into ME/CFS pathophysiology, offering potential biomarkers for diagnosis and targets for therapeutic intervention.

## 5. Conclusions

Mitochondrial dysfunction is a critical factor in the pathophysiology of ME/CFS, driving reduced ATP production rates, impaired oxidative phosphorylation, and redox imbalance, which collectively exacerbate fatigue, exercise intolerance, and other debilitating symptoms. While evidence for immune senescence in patients with ME/CFS remains inconclusive, the interplay between mitochondrial dysfunction and immune dysregulation, involving immune exhaustion, chronic inflammation, and oxidative stress, underscores the need for further research to clarify these mechanisms and identify potential therapeutic targets.

The findings outlined in this review provide a comprehensive overview of the current knowledge on mitochondrial and immune dysfunction in ME/CFS, offering valuable insights into disease pathophysiology and potential biomarker discovery. However, conflicting results across studies, methodological variability, differences in diagnostic criteria, and variations in patient cohorts underscore the need for standardized research approaches. Given the significant heterogeneity among patients—contributing to inconsistent findings—future studies should prioritize large, well-characterized cohorts and adopt longitudinal study designs. The metabolic–immune interplay presents a promising avenue for advancing our understanding of ME/CFS. However, key knowledge gaps remain, emphasizing the urgent need for further research.

## Figures and Tables

**Figure 1 biomolecules-15-00357-f001:**
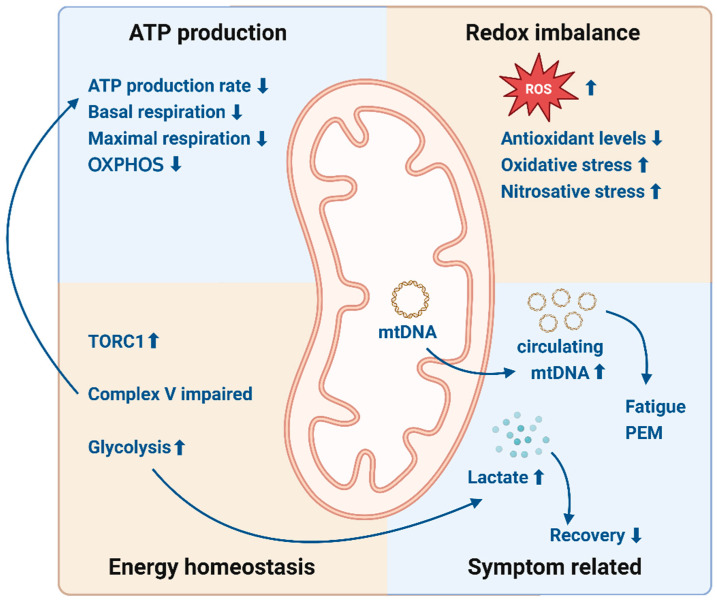
Mitochondrial dysfunction in ME/CFS. Reduced ATP levels, basal respiration, maximal respiration, and OXPHOS, alongside impaired Complex V and a switch to glycolysis as a compensatory response are indicated in patients with ME/CFS. Elevated ROS levels and decreased antioxidant levels contribute to oxidative and nitrosative stress, further exacerbating the imbalance. Fatigue, PEM, and impaired recovery are symptoms linked to these metabolic and mitochondrial dysfunctions. Elevated levels of exosome-associated mtDNA in serum are correlated with the severity of ME/CFS symptoms. Abbreviations: ATP = adenosine triphosphate, OXPHOS = oxidative phosphorylation, ROS = reactive oxygen species, TORC1 = Target of Rapamycin Complex I, PEM = post-exertional malaise, mtDNA = mitochondrial DNA, ↑ = increased, and ↓ = decreased. Created in BioRender (https://app.biorender.com/, accessed on 21 February 2025). Van campenhout, J. (2025) https://BioRender.com/p62w552, accessed on 21 February 2025.

**Figure 2 biomolecules-15-00357-f002:**
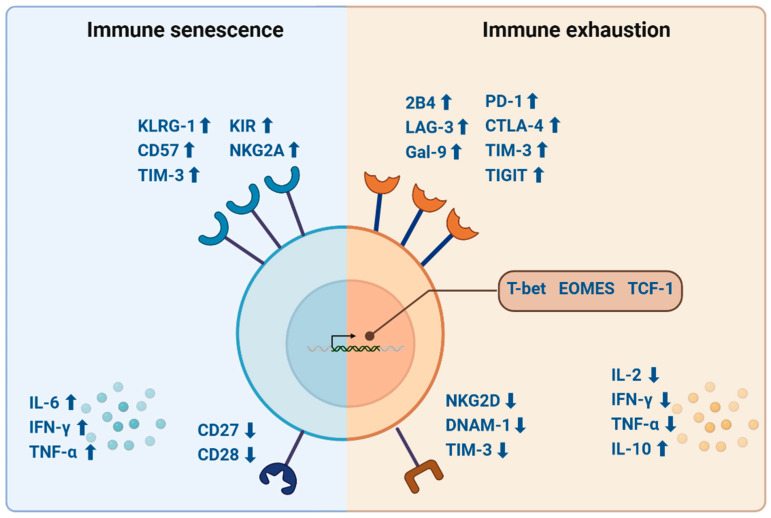
Characterization of immune senescence and immune exhaustion. The senescent and exhausted immune states can be characterized by their expression of inhibitory and activating cell surface receptors, levels of proinflammatory cytokines, and transcriptional changes. Abbreviations: 2B4 = also known as CD244; CD27 = cluster of differentiation 27; CD28 = cluster of differentiation 28; CD57 = cluster of differentiation 57; CTLA-4 = cytotoxic T-lymphocyte-associated antigen 4; DNAM-1= DNAX accessory molecule 1; EOMES = eomesodermin; Gal-9 = galectin-9; IFN-γ = interferon gamma; IL-2 = interleukin 2; IL-6 = interleukin 6; IL-10= interleukin 10; KIR= killer cell immunoglobulin-like receptors; KLRG-1 = killer cell lectin-like receptor G1; LAG-3 = lymphocyte-activation gene 3; NKG2A = natural killer group 2A; NKG2D= natural killer group 2D; PD-1 = programmed death 1; T-bet = T-box expressed in T cells; TCF-1 = T cell factor 1; TIGIT = T-cell immunoreceptor with Ig and ITIM domains; TIM-3 = T cell immunoglobulin and mucin domain-containing molecule 3; TNF-α = tumour necrosis factor alpha; ↑ = increased; ↓ = decreased. Created in BioRender (https://app.biorender.com/, accessed on 24 January 2025). Buntinx, Y. (2025) https://BioRender.com/h04l833, accessed on 24 January 2025.

**Figure 3 biomolecules-15-00357-f003:**
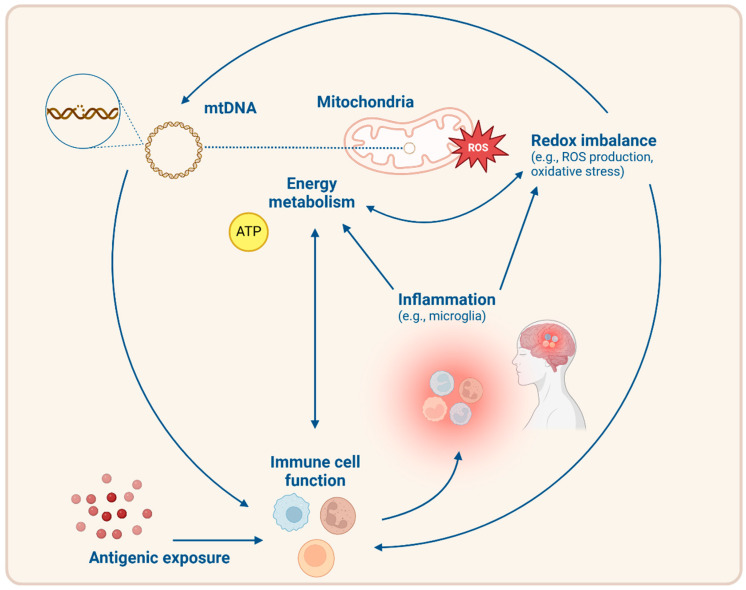
Schematic representation of the bidirectional feedback loop between mitochondrial and immune dysfunction in patients with ME/CFS. Mitochondrial dysfunction leads to redox imbalance, including increased ROS production and oxidative stress, which can damage mtDNA. Circulating mtDNA acts as a damage-associated molecular pattern (DAMP), triggering immune activation and inflammation, including inflammation of microglia in the central nervous system. Chronic antigenic exposure and metabolic disturbances further impair immune cell function, potentially contributing to immune senescence and exhaustion. These immune states exacerbate mitochondrial stress, reinforcing energy metabolism deficits and thereby perpetuating the cycle of mitochondrial and immune dysfunction. Abbreviations: ATP = adenosine triphosphate, mtDNA = mitochondrial DNA, and ROS = reactive oxygen species. Created in BioRender (https://app.biorender.com/, accessed on 24 January 2025). Buntinx, Y. (2025) https://BioRender.com/w73q827, 24 January 2025.

**Table 1 biomolecules-15-00357-t001:** Glossary.

Useful Terms	Description
Terms related to energy metabolism	
Aerobic energy metabolism	The process of producing energy in cells through the breakdown of nutrients in the presence of oxygen, primarily occurring in the mitochondria and involving oxidative phosphorylation.
Anaerobic energy metabolism	Energy production in cells without oxygen, mainly through glycolysis, resulting in the formation of lactate as a byproduct.
Glycolysis	A metabolic pathway that breaks down glucose into pyruvate, generating a small amount of ATP and NADH, and occurs in the cytoplasm.
Oxidative phosphorylation	A process of generating ATP in the mitochondria that uses a series of mitochondrial complex enzymes (Complexes I-IV) in the electron transport chain to transfer electrons and pump protons, with oxygen acting as the final electron acceptor.
Basal respiration	The minimal level of oxygen consumption by cells needed to maintain basic metabolic functions at rest.
Maximal respiration	The highest rate of oxygen consumption a cell can achieve under conditions of maximal electron transport chain activity, often stimulated by uncouplers.
Spare respiratory capacity	The difference between basal respiration and maximal respiration, representing a cell’s ability to respond to increased energy demands.
Oxidative stress	An imbalance between the production of reactive oxygen species and the body’s ability to detoxify them or repair the resulting damage.
Reactive oxygen species	Chemically reactive molecules containing oxygen that are byproducts of energy metabolism, primarily generated in the mitochondria during processes like oxidative phosphorylation, and can damage cells if not regulated.
Antioxidant capacity	The ability of a cell or organism to neutralize reactive oxygen species and prevent oxidative damage using antioxidants.
Nitrosative stress	Cellular stress linked to energy metabolism, caused by an excess of reactive nitrogen species, which are often associated with mitochondrial dysfunction and disruptions in metabolic processes.
Terms related to the immune system	
Immune exhaustion	A state in which immune cells, particularly T cells, lose their ability to function effectively due to chronic activation.
Immune senescence	The gradual decline in immune function associated with ageing, characterized by reduced immune cell activity and impaired ability to respond to new infections.
T cell	A type of white blood cell essential for adaptive immunity, responsible for killing infected or cancerous cells and regulating immune responses.
B cell	A type of white blood cell involved in adaptive immunity, primarily responsible for producing antibodies to neutralize pathogens.
Natural killer cell	A type of immune cell that is part of the innate immune system, capable of destroying virus-infected and cancerous cells without prior activation.
Cytokines	Small signalling proteins released by immune cells that mediate and regulate immunity and inflammation.

## Data Availability

Not applicable.

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
