# Peer review of "Unravelling the Connection Between Energy Metabolism and Immune Senescence/Exhaustion in Patients with Myalgic Encephalomyelitis/Chronic Fatigue Syndrome"

_biomolecules, 2025, doi:10.3390/biom15030357_

Round 1
Reviewer 1 Report
Comments and Suggestions for Authors
The present review describes the role of energy metabolism in Myalgic Encephalomyelitis/Chronic Fatigue Syndrome (ME/CFS). Here are some minor comments to increase the impact for the resaders
- Lines 112–113: The authors mention that the ATP Profile test provides insight into mitochondrial function. Please include statistical data and specify the sample used for ATP profiling between ME/CFS patients and healthy controls (HC), along with proper referencing. Additionally, are there any reports on glycolytic function in ME/CFS? If so, including that information would enhance the impact of this section.
- Line 17: The authors discuss Carnitine dysfunction. It would be scientifically more accurate to use the term Carnitine levels and mitochondrial dysfunction instead of Carnitine dysfunction.
- Figure 1: What is the purpose of showing mtDNA in the figure, as there is no discussion about it in the text? Instead, it would be more informative to illustrate the electron transport chain or provide additional details on the role of mtDNA in ME/CFS within the figure.
- Figure 3: Please clarify in the figure legend what the authors intend to convey with this figure.
- Since the review highlights the current understanding of cellular bioenergetics in ME/CFS, it would be valuable to include a paragraph discussing new findings, such as the link between the metabolome and the epigenome in this disease. Additionally, if there are any reports on histone lactylation or histone and histone acetylation in this context,including those references would further strengthen the discussion.
- In the conclusion it would be helpful to emphasize future implications, identify gaps in existing studies, and suggest potential directions for further research.
Reviewer 2 Report
Comments and Suggestions for Authors
This manuscript reviews some of the literature on energy metabolism generally and more specifically in immune cells. I would have organized the review differently, but that is just a matter of preference and what they have done is acceptable. Sometimes it was difficult to know whether the authors were reviewing phenomena in general or specifically about ME/CFS, though they attempted to differentiate the topics with headings.
There has been almost no work indicating immune senescence, but there are papers implicating immune exhaustion. The authors overlook the paper by Walitt et al which provide some, albeit a bit weak, evidence about immune exhaustion. https://pmc.ncbi.nlm.nih.gov/articles/PMC10881493/
In the metabolic-immune link section, the authors fail to describe this paper, which is the only one I know about that looked at fatty acid metabolism in ME/CFS. https://pubmed.ncbi.nlm.nih.gov/36768336/
Is reference 47 the one the authors mean to cite on line 479?
